# Application of Airborne Microorganism Indexes in Offices, Gyms, and Libraries

**Pietro Grisoli** [1,*] , **Marco Albertoni** [1] **and Marinella Rodolfi** [2]

1   Department of Drug Sciences, Laboratory of Microbiology, University of Pavia, 27100 Pavia, Italy; marco.albertoni@unipv.it

2   Department of Earth and Environmental Sciences, Mycology Section, University of Pavia, 27100 Pavia, Italy; marinella.rodolfi@unipv.it

*   Correspondence: pietro.grisoli@unipv.it; Tel.: +39-0382-987397

**Abstract:** The determination of microbiological air quality in sporting and working environments requires the quantification of airborne microbial contamination. The number and types of microorganisms, detected in a specific site, offer a useful index for air quality valuation. An assessment of contamination levels was carried out using three evaluation indices for microbiological pollution: the global index of microbiological contamination per cubic meter ($GIMC/m^3$), the index of mesophilic bacterial contamination (IMC), and the amplification index (AI). These indices have the advantage of considering several concomitant factors in the formation of a microbial aerosol. They may also detect the malfunction of an air treatment system due to the increase of microbes in aeraulic ducts, or inside a building compared to the outdoor environment. In addition, they highlight the low efficiency of a ventilation system due to the excessive number of people inside a building or to insufficient air renewal. This study quantified the levels of microorganisms present in the air in different places such as offices, gyms, and libraries. The air contamination was always higher in gyms that in the other places. All examined environments are in Northern Italy.

**Keywords:** airborne microorganisms; bacteria; fungi; gyms; indoor air quality; libraries; offices

---

## 1. Introduction

The evaluation of microbiological air contamination is an important aspect of applied microbiology and industrial hygiene [1–3]. In Italy, several legislative decrees, in particular the Legislative Decree 81/2008, impose that employers are responsible for the assessment of risks of biological origin arising from the activities in the workplace; moreover, they must adopt suitable measures so that the air in the workplace be healthy [4]. Therefore, the realization of measuring methods of airborne contamination is a matter of prime importance in the most complex procedure of risk evaluation that must involve, necessarily, different scientific skills [5–8]. Air contamination, both in confined environments and outdoors, can be caused not only by chemical agents but also by biological sources such as pollen, dust, mites, insects, pet allergens, bacteria, fungi, and viruses [9]. The monitoring of bioaerosols represents an important tool both for assessing the risk in the working environment [10] and for the evaluation of the environmental impact of certain activities that take place outdoors, such as biopurification or waste disposal [11]. Several investigations have been directed to study the disease called Sick Building Syndrome (SBS), which is characterized by varied and non-specific symptoms observed in workers employed in confined environments [12–14]. The World Health Organization has defined SBS as "an increase in frequency in the occupants of non-industrial buildings, of not specific acute symptoms (irritation of eyes, nose, throat, headache, fatigue, nausea) that improve when building is left" [15].

The American Conference of Governmental Industrial Hygienists (ACGIH) does not suggest threshold limit values (TLV) for the environmental concentration of biological agents, because the existing information does not allow to establish a scientifically acceptable dose–response relationship [16].

However, the assessment of culturable microbial agents is currently the easiest and most practical measure to determine changes in the air quality of confined workplaces. The purpose of these controls may be to set guide values, technologically attainable by adopting containment measures of contamination established on the basis of repeated samplings in specific locations. With this aim, this study examined structures destined for different types of work and recreational activity such as offices, gyms, and libraries, measuring air-diffused contamination to identify and apply indices useful for the microbiological classification of the air quality [17].

The indoor air quality was evaluated by applying the following indices: the global index of microbiological contamination per cubic meter (GIMC/m$^3$), the index of mesophilic bacterial contamination (IMC), and the amplification index (AI). The GIMC considers different microbial types and emphasizes their ability to grow in a wide temperature range. GIMC may be a simple method to evaluate a potential biological risk in indoor and outdoor environments and to monitor sources of microbiological contamination. The IMC reveals the presence of obligated mesophilic bacteria, organisms of probable human origin in the indoor air. This index represents a practical instrument to underline bacterial growth caused by hypoventilation and overcrowding. The AI is important as it reveals microbial pollution in indoor ventilation systems. These indices showed their applicability in other types of environments [18,19].

## 2. Materials and Methods

### 2.1. Sampling Method

Environmental monitoring was performed in Northern Italy in 10 diverse offices, gyms, and libraries situated in buildings equipped with a ventilation system able to work in the following modes: heating, air conditioning, and simple ventilation. The offices are placed in bank edifices, the gyms in fitness centers, while the libraries belong to university buildings. The samplings were realized every month for one year, inside the various environments, during normal people activity, and outside the building. Quantitative data were determined in triplicate by means of an orthogonal impact Microflow Air Sampler (AQUARIA, Lacchiarella, Italy), kept 1.5 m above ground level, in the center of the room for offices, in the middle of the weight-training room for gyms, and in the reading rooms for libraries. Air samplers worked at a fixed speed of 1.5 Ls$^{-1}$, collecting a volume of 200 L. For the different types of environments, the number of persons (n. p.) and the air speed (m/s) were recorded, so that in three sampling periods, the following average values were obtained: offices (n. p. 4.7; 0.03 m/s), gyms (n. p. 21.3; 0.060 m/s), and libraries (n. p. 16.2; 0.05 m/s).

### 2.2. Microorganisms Assessed

Bacteria were collected using Tryptone Soya Agar (TSA, Oxoid, Basingstoke, UK), and the cultures were incubated at 37 °C for 48 h for mesophilic bacteria and at 20 °C for 6 days for psychrophilic bacteria. Fungi were collected on Sabouraud Dextrose Agar (SAB Oxoid, Basingstoke, UK), and the cultures were incubated at 20 °C for 6 days. The outdoor air quality used as control was analyzed following the same criteria. All total microbial counts are indicated as the number of colony-forming units per cubic meter of air (CFU/m$^3$), calculated as an average of three determinations from three samples collected serially. The assessment of microbial contamination was effected by using bacterial and fungal counts, and the following indexes were calculated: GIMC per cubic meter (GIMC/m$^3$), which is the sum of the values of the total microbial counts determined for mesophilic bacteria, psychrophilic bacteria, and fungi in all sampled areas; IMC, derived from the ratio between the values of CFU per cubic meter measured for mesophilic and psychrophilic bacteria at the same sampling

point; AI, resulting from the ratio between the GIMC/m$^3$ values measured inside the building and those measured outdoor.

*2.3. Data Analysis*

The air microbial contamination values were expressed as colony-forming units (CFUs), and the limit of quantification was 1 CFU/m$^3$. The number of CFUs for contact plate after appropriate incubation was corrected using the positive-hole correction table provided by the supplier. Statistical analysis of the data was performed by comparing the results obtained in the different environments by means of analysis of variance one-way (post hoc test) values transformed into logarithms (natural log). The analyses were conducted using Prism 3.0. The significance level was $p < 0.05$.

## 3. Results

Table 1 shows the results of microbiological contamination of 10 different offices located in buildings with centralized systems for ventilation. The mean values were higher for mesophilic bacteria during the air conditioning phase compared to heating ($p = 0.034$) and during simple ventilation compared to heating ($p = 0.023$). The contamination of psychrophilic bacteria was greater during simple ventilation compared to air conditioning ($p = 0.016$) and to heating ($p = 0.032$). The mean values of CFU/m$^3$ for the fungi were not significantly different during the different periods of air treatment. Although the average contamination was low, there were high maximum values of CFU/m$^3$ for mesophilic bacteria in the periods of air conditioning and simple ventilation and for psychrophilic bacteria during simple ventilation.

**Table 1.** Total microbial concentrations measured in the offices during heating, air conditioning, and simple ventilation.

| Functioning Modes | N | Mesophilic Bacteria CFU/m$^3$ | | Psychrophilic Bacteria CFU/m$^3$ | | Fungal Count CFU/m$^3$ | |
|---|---|---|---|---|---|---|---|
| | | M ± SD | Min–max | M ± SD | Min–max | M ± SD | Min–max |
| Heating | 10 | 191.20 ± 144.53 | 8–450 | 176.70 ± 117.21 | 7–366 | 136.00 ± 32.80 | 22–654 |
| Simple ventilation | 10 | 667.60 ± 1602.60 | 6–5200 | 761.90 ± 1519.7 | 10–5000 | 131.50 ± 72.30 | 5–190 |
| Air Conditioning | 10 | 1089.70 ± 2392.20 | 19–7800 | 495.60 ± 307.20 | 76–860 | 119.60 ± 48.84 | 28–199 |

The number of samples (N), mean (M), standard deviation (SD), and range of values are indicated for each sampling period. CFU: colony-forming units.

The calculation of GIMC/m$^3$ confirmed the presence of higher values of contamination during conditioning ($p = 0.01$) and simple ventilation ($p = 0.01$) compared to heating. The average value of IMC during heating amounted to 7.30 and was higher than the values of the other modes of operation, corresponding to 0.80 during the simple ventilation ($p = 0.04$) and 1.60 during air conditioning (Table 2).

**Table 2.** Global index of microbial contamination per cubic meter (GIMC/m$^3$) and index of mesophilic bacterial contamination (IMC) measured in offices during heating, air conditionining and simple ventilation.

| Functioning Modes | N | GIMC/m$^3$ | | IMC | |
|---|---|---|---|---|---|
| | | M ± SD | Min–max | M ± SD | Min–max |
| Heating | 10 | 503.90 ± 282.22 | 124–1111 | 7.30 ± 20.03 | 0.20–64.30 |
| Simple ventilation | 10 | 1561.00 ± 3154.80 | 21–10,450 | 0.80 ± 0.40 | 0.10–1.30 |
| Air Conditioning | 10 | 1704.90 ± 2537.57 | 229–8720 | 1.60 ± 2.67 | 0.02–9.10 |

The number of samples (N), mean (M), standard deviation (SD), the range of values are indicated for each sampling period.

The data regarding the microbiological contamination found in 10 different gyms with centralized ventilation systems are summarized in Table 3. The results show highest mean values for mycetic contamination during the air conditioning phase compared to simple ventilation ($p = 0.04$) and heating ($p = 0.01$). As regards bacteria, the average contamination values for psychrophilic bacteria were

superior to those for mesophilic bacteria in the three air treatment modalities and were significantly different during conditioning ($p = 0.02$).

**Table 3.** Total microbial concentrations measured in gyms during heating, air conditioning, and simple ventilation.

| Functioning Modes | N | Mesophilic Bacteria CFU/m$^3$ | | Psychrophilic Bacteria CFU/m$^3$ | | Fungal Count CFU/m$^3$ | |
|---|---|---|---|---|---|---|---|
| | | M ± SD | Min–max | M ± SD. | Min–max | M ± SD | Min–max |
| Heating | 10 | 1096.60 ± 924.40 | 140–2850 | 1446.40 ± 1356.82 | 180–4650 | 187.50 ± 167.02 | 20–568 |
| Simple ventilation | 10 | 393.00 ± 257.00 | 120–980 | 819.00 ± 432.40 | 330–1800 | 1208.90 ± 1459.60 | 90–4848 |
| Air Conditioning | 10 | 666.50 ± 477.76 | 200–1800 | 1509.00 ± 1354.02 | 440–4400 | 2430.90 ± 3678.31 | 8–10,848 |

The number of samples (N), mean (M), standard deviation (SD), and range of values are indicated for each sampling period.

There was a notable increase in fungal count when the central heating was switched off. In fact, the highest maximum values of CFU/m$^3$ were found for fungi during simple ventilation and conditioning, in contrast to what observed for bacteria ($p = 0.016$).

Table 4 shows the results of microbial contamination based on the GIMC/m$^3$ and on the IMC measured in the gyms. Because of the elevated fungal count observed in this period, the mean of GIMC/m$^3$ was higher during air conditioning ($p = 0.019$), with a maximum value of 15,248. The mean of the IMC values was always <1, which implies that the counts of the psychrophilic bacteria were almost always higher than the counts of the mesophilic bacteria; however, no significant differences in the index were observed when central heating was on or off.

**Table 4.** GIMC/m$^3$ and IMC measured in gyms during heating, air conditioning, and simple ventilation.

| Functioning Modes | N | GIMC/m$^3$ | | IMC | |
|---|---|---|---|---|---|
| | | M ± SD | Min–max | M ± SD | Min–max |
| Heating | 10 | 2703.50 ± 2144.84 | 1160–7090 | 0.90 ± 0.74 | 0.12–2.40 |
| Simple ventilation | 10 | 2420.90 ± 1645.70 | 910–6548 | 0.50 ± 0.30 | 0.17–1.20 |
| Air Conditioning | 10 | 4606.40 ± 4428.70 | 1480–15,248 | 0.60 ± 0.39 | 0.20–1.40 |

The number of samples (N), mean (M), standard deviation (SD), and range of values are indicated for each sampling period.

Similar sampling and monitoring methods were adopted for the evaluation of the air quality of libraries. The average levels of contamination detected during air sampling showed small oscillations with very restricted values of CFU/m$^3$ both for bacterial loads (mesophilic and psychrophilic) and for fungi. The results of airborne bacteria and fungi recovered from the libraries showed that the CFU/m$^3$ values were higher during heating than during simple ventilation or air conditioning. However, no significant differences in the concentrations of all microbial counts were evidenced in the three functioning modes of the ventilation system (Table 5).

**Table 5.** Total microbial concentrations in libraries during heating, air conditioning, and simple ventilation.

| Functioning Modes | N | Mesophilic Bacteria CFU/m$^3$ | | Psychrophilic Bacteria CFU/m$^3$ | | Fungal Count CFU/m$^3$ | |
|---|---|---|---|---|---|---|---|
| | | M ± SD | Min–max | M ± SD | Min–max | M ± SD | Min–max |
| Heating | 10 | 171.20 ± 72.43 | 112–320 | 301.10 ± 237.26 | 125–725 | 123.90 ± 54.52 | 62–220 |
| Simple ventilation | 10 | 65.10 ± 29.40 | 38–110 | 152.40 ± 90.70 | 81–380 | 66.00 ± 31.70 | 32–110 |
| Air Conditioning | 10 | 71.80 ± 30.50 | 30–120 | 200.60 ± 127.65 | 97–400 | 76.60 ± 36.55 | 35–120 |

The number of samples (N), mean (M), standard deviation (SD), and range of values are indicated for each sampling period.

In addition, the transformation of total microbial count values into the indices evidenced a general reduction of the GIMC/m$^3$ values and of the IMC values in air conditioning compared to heating, but the statistical analysis showed no significant differences. The IMC index assumed values lower

than 1 and showed that air contamination was mainly due to environmental bacteria. All values were limited to a small range in every period of the monitoring campaign (Table 6).

**Table 6.** GIMC/m$^3$ and IMC measured in libraries during heating, air conditioning, and simple ventilation.

| Functioning Modes | N | GIMC/m$^3$ | | IMC | |
|---|---|---|---|---|---|
| | | M $\pm$ SD | Min–max | M $\pm$ SD | Min–max |
| Heating | 10 | 596.20 $\pm$ 335.08 | 300–1250 | 0.70 $\pm$ 0.22 | 0.40–0.90 |
| Simple ventilation | 10 | 283.50 $\pm$ 131.70 | 153–578 | 0.50 $\pm$ 0.20 | 0.25–0.70 |
| Air Conditioning | 10 | 349.00 $\pm$ 176.43 | 182–652.50 | 0.40 $\pm$ 0.18 | 0.20–0.60 |

The number of samples (N), mean (M), standard deviation (SD), and range of values are indicated for each sampling period.

In indoor environments, microbiological air contamination can also be described by the Amplification Index of microbial contamination (AI). To calculate this index, it was necessary to use the GIMC values measured outside the buildings in the different sampling periods (Table 7). These values were significantly different in the three sampling points; in particular, the widest variations were detectable outside gyms in the air conditioning period ($p < 0.0001$).

**Table 7.** GIMC/m$^3$ measured outside gyms, libraries, and offices during heating, air conditioning, and simple ventilation periods.

| Sampling Points | N | Heating GIMC/m$^3$ | | Simple Ventilation GIMC/m$^3$ | | Air Conditioning GIMC/m$^3$ | |
|---|---|---|---|---|---|---|---|
| | | M $\pm$ SD | Min–max | M $\pm$ SD | Min–max | M $\pm$ SD | Min–max |
| Gyms- outdoor | 10 | 714.40 $\pm$ 294.00 | 328–1268 | 703.10 $\pm$ 226.40 | 484–1236 | 3586.10 $\pm$ 2326.42 | 1325–9240 |
| Libraries outdoor | 10 | 410.65 $\pm$ 187.81 | 266–880 | 596.55 $\pm$ 167.19 | 359–892 | 508.40 $\pm$ 141,01 | 364–830 |
| Offices-outdoor | 10 | 421.70 $\pm$ 179.62 | 248–890 | 619.90 $\pm$ 169.30 | 392–933 | 703.80 $\pm$ 226.20 | 485–1068 |

The number of samples (N), mean (M), standard deviation (SD), and range of values are indicated for each sampling period.

AI is determined by calculating the ratio between the GIMC/m$^3$ values measured inside a building and those measured outside. This index is greater than 1 when the microbial contamination inside a building is higher than the outdoor contamination. In most cases, the values of microbial contamination of indoor air are higher than the outdoor values. In the heating phase, AI showed a worsening of air contamination especially in gyms compared the other environments considered (3.7). On the contrary, during the conditioning period, the highest values were recorded in offices (3.0) (Figure 1).

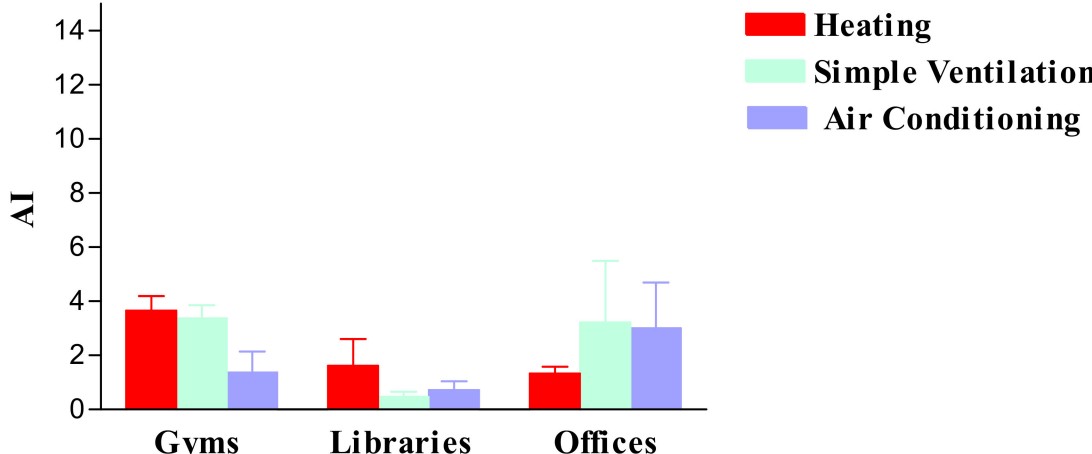

**Figure 1.** Amplification Index of microbial contamination (AI) measured in the three types of environments during heating, air conditioning, and simple ventilation.

## 4. Discussion

It can be reasonably assumed that the microflora existing in recreational and working indoor environments has a concentration lower than or equal to the external one detected in the same location and in the same climatic conditions [20,21]. In buildings equipped with centralized ventilation systems there is a reduction of microbial contamination compared to the outside when air filtration and treatment systems are maintained in appropriate conditions [22]. The air inside buildings may be contaminated by the growth of microorganisms on floors, walls, aeraulic plants or because of inadequate air changes [23]. The ventilation systems can have very different structural characteristics, ranging from the simple aspiration of exhaust air to integrated heating, cooling, and humidification. It is, therefore, clear that the air introduced from these systems can be considered as a matrix whose microbiological quality can be evaluated and classified, as for any other product that may have effects on human health. In addition, the activities carried out in a confined environment may be responsible for the spread of microorganisms.

Dissenting opinions have been expressed on the possibility to monitor microbiological environmental contamination in workplaces. Information on the dose–response relationship related to the exposure to microorganisms is still not available today. The ACGIH does not indicate TLV for biological agents. Furthermore, different analytical methods do not allow to recover and identify all the microorganisms present in the air. In fact, the percentage of viable, culturable bacteria recovered with normal sampling systems oscillates in a range of values that varies between 0.1% and 10% of the total bacteria present in the air [24–26]. The presence of microorganisms should also be sought in working environments because of their toxigenic potential and the possibility of spreading cell fragments and volatile organic compounds into the environment. The lack of viable, culturable cells does not necessarily indicate a healthy environment. Some studies, carried out in diversified working environments, report environmental contamination values referable to the number of viable cells belonging to a single class of microorganisms [27–29]. For example, a microbiological classification of the air quality in non-industrial environments and homes considers the contamination from bacteria that develop at 20–25 °C. In this classification, for non-industrial environments, a level <100 CFU/m$^3$ corresponds to the category of low contamination, while a value >2000 CFU/m$^3$ corresponds to the category of very high contamination [30]. It is clear that such assessment is incapable not only to describe all the factors that determine the accumulation and spread of microorganisms in the environment but also to identify possible risks for workers.

In this research, GIMC was used in order to include in the same data several categories of microorganisms. This index considers microorganisms proliferating at different temperatures, such as mesophilic and psychrophilic bacteria, and fungi capable of adapting to different types of environments. These characteristics make this quantitative measure of microbiological contamination significant and allow to evaluate the salubrity of a work environment; in fact, it considers microorganisms that can develop in a wide range of temperatures, including ambient temperatures, typical of saprophytic life, and 37 °C, which is the temperature of development of pathogens. This index is particularly useful because it is able to highlight even anomalous situations in indoor environmental microbiological contamination: in fact, in the offices we analyzed, the mesophilic bacteria during the conditioning phase were higher than the psychrophilic ones. This value represents an exception because it is usually reasonable to assume that psychrophilic bacteria, which might grow well in air conditioning systems, are more numerous than mesophilic bacteria during the cooling season, as shown in the results for gyms and libraries buildings and in previous researches [22]. In particular, GIMC attributes importance to bacteria that can proliferate in a wide range of temperatures. It is true that the two incubation temperatures (20 °C and 37 °C) do not differentiate the two categories of bacteria completely. Nevertheless, it is useful to determine the two total counts during environmental monitoring because they have different significances and allow a more complete evaluation of airborne bacterial contamination. In fact, the purpose of the index is to provide a measure of biological risk and to aggregate several "environmental indicators". IMC is mainly an index of anthropic contamination;

it highlights the share of mesophilic bacteria in the microbial population examined. This index derives from the ratio between the value of $CFU/m^3$ at 37 °C and that at 20 °C. This value, determined outdoor, is always very close to 1, whereas in indoor environments, it may be higher, also depending on the number of people present. In fact, mesophilic bacteria derive from the normal bacterial flora of humans and can therefore constitute the predominant population in confined environments, as already verified in other works [9]. Finally, the amplification index is fundamental to detect the accumulation and proliferation of microorganisms in ventilation systems or in buildings. AI describes global indoor aerial modification. Generally, there are no relations between indoor and outdoor fungal and bacterial counts. However, it is important to judge Indoor Air Quality (IAQ) not only by the measurement of single parameters, but also using the global value of microbial contamination. High AI values may only be indicative of IAQ deterioration when caused by high total fungal and bacterial counts [10,22]. Moreover, AI is essential to determine the environmental impact of outdoor work activities with potential spread of pathogenic and non-pathogenic microorganisms; it must be calculated by referring to the contamination detected in a control point that must provide the background value [18]. AI and the other two indices could provide criteria to evaluate indoor environmental quality more uniformly [31]. In fact, these indices have different advantages: they consider several factors which contribute to the development of different types of microorganisms (determination of GIMC), they may reveal an inadequate functioning of the air conditioning system due to an excessive number of people in a building or insufficient ventilation (IMC determination), and they may suggest a malfunction of the air system due to the increase of microbial contamination in the ventilation ducts or in the building compared to the outside environment (AI measures). The utilization of these indices can enable the determination of threshold values as a function of the structures (buildings) analyzed. In previous research in other environments, such as university classrooms, a reference value of $GIMC/m^3 = 1000$ was connected to a correct maintenance of the aeraulic system [19]. In this work, the measured GIMC values were higher than 1000, with the exception of libraries that presented very low contamination. In particular, both offices and gyms had the highest values of pollution during the conditioning phase, corresponding to $GMIC/m^3$ of 1704.90 and 4606.40, respectively. GIMC exceeding 1000 does not necessarily indicate a health risk; however, it is appropriate to analyze the contamination levels through the calculation of IMC and AI. During air conditioning in the offices, IMC and AI showed that mesophilic bacteria underwent a real amplification due to few air changes or the accumulation of microorganisms in the air system. In fact, as reported by other authors, variations and fluctuations in indoor humidity and temperature have significant effects on microbial diffusion and growth [32]. For gyms, on the contrary, the GIMC values were higher in summer because the proportion of mesophilic bacteria decreased passing from the heating phase to the conditioning phase, but the portion of psychrophilic bacteria and mycetes increased consistently, as evidenced by the IMC.

It is important to note that the rise of the microbial load inside the gyms coincided with an increase of the microorganisms outdoors, as shown by the calculation of AI. Therefore, the presence of a greater microbial contamination during summer could depend on the incorrect functioning or poor controls of the ventilation systems, which resulted in the accumulation of microorganisms from the outside [33].

In conclusion, our research results show that the microbial contamination changes depending on the environment analyzed and highlight the easy applicability of the proposed indices. GIMC, IMC, and AI complete the information provided by the single classes of airborne microorganisms and, together, they allow the analysis of indoor air quality. On the basis of sufficient data, these indices classify environments in function of their microbiological contamination and identify guide values to be adopted for routine monitoring and for the implementation of containment and remediation measures.

**Author Contributions:** Conceptualization, P.G.; methodology, P.G.; formal analysis, P.G., M.R.; investigation, P.G., M.A., M.R.; funding acquisition, P.G., M.A.; writing—original draft preparation, P.G. and M.R.

**Funding:** This work was financially supported by a research fund of the Laboratory of Microbiology, Department of Drug Sciences, University of Pavia.

**Conflicts of Interest:** The authors declare no conflict of interest.

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
