# Peer review of "Application of Airborne Microorganism Indexes in Offices, Gyms, and Libraries"

_applsci, doi:10.3390/app9061101_

Round 1
Reviewer 1 Report
This study introduces the application case of the index indicating airborne microbial contamination to three types of buildings. The subject of this study is interesting and worth investigating, but I recommend the authors answer some questions and revise the manuscript.
- As for GIMC, the sources of bacteria and fungi are usually not same and so it is hard to evaluate the airborne microbial contamination by summing up the different kind of microorganism. The dominant sources of airborne bacteria are human activities and sometimes contaminated air systems. And outdoor concentration of bacteria is mostly not higher than indoor. However, the sources of airborne fungi are outdoor and fungal contamination on indoor surfaces, and so high airborne fungal concentration could be from indoor or/and outdoor. It means that indoor airborne fungal spore can increase depending on where the building is located. Therefore, I think it is very hard to find out the sources and evaluate the contamination from GIMC.
- The information, such as occupancy ratios when measured, air filters of mechanical systems and ventilation rate, can help authors and readers to understand and analyze the measuring results.
- The reasons why the measuring results in three type of buildings were varied by operating mode of mechanical systems were not well explained. For example, the psychrophilic bacteria concentration was lower than mesophilic one during air conditioning in offices. However, we can assume that psychrophilic bacteria, which might grow well in air conditioning systems, are more than mesophilic one during cooling season, as shown in the results for gyms and libraries buildings.
- I am not sure if the duplicated species growing on the same agar with different incubating temperatures do not reduce reliability of GIMC.
Author Response
Reviewer 1
Comments and Suggestions for Authors
This study introduces the application case of the index indicating airborne microbial contamination to three types of buildings. The subject of this study is interesting and worth investigating, but I recommend the authors answer some questions and revise the manuscript
We did our best to sustain all the objection of the Reviewer. The aim of our research is to apply this method to various occupational environments to verify its efficacy and to provide data useful for constructing reference values to evaluate IAQ.
- As for GIMC, the sources of bacteria and fungi are usually not same and so it is hard to evaluate the airborne microbial contamination by summing up the different kind of microorganism. The dominant sources of airborne bacteria are human activities and sometimes contaminated air systems. And outdoor concentration of bacteria is mostly not higher than indoor. However, the sources of airborne fungi are outdoor and fungal contamination on indoor surfaces, and so high airborne fungal concentration could be from indoor or/and outdoor. It means that indoor airborne fungal spore can increase depending on where the building is located. Therefore, I think it is very hard to find out the sources and evaluate the contamination from GIMC.
We agree with the Reviewer on the fact that there are no relations between indoor and outdoor fungal and bacterial counts. However, the determination of total bacteria and fungal counts seems a more practical approach in environmental monitoring, but should not exclude other specific investigations. Though open to criticism, GIMC, IMC and AI indices may represent simple and useful tools for the measurement and evaluation of biological risks. The proposed indices seek to highlight aerial microbiological contamination, both quantitatively and qualitatively. In particular, GIMC attributes an important role to bacteria which can proliferate in wide temperature ranges. We are aware that the two incubation temperatures (20°C and 37°C) do not differentiate the two categories of bacteria completely. Nevertheless, we believe that it is useful to determine the two total counts in environmental monitoring because they have different significances and allow a more complete evaluation of airborne bacterial contamination. Moreover, considering fungal count, GIMC may provide a measure of biological risk because it aggregates several “environmental indicators”.
- The information, such as occupancy ratios when measured, air filters of mechanical systems and ventilation rate, can help authors and readers to understand and analyze the measuring results.
Done.
-The reasons why the measuring results in three type of buildings were varied by operating mode of mechanical systems were not well explained. For example, the psychrophilic bacteria concentration was lower than mesophilic one during air conditioning in offices. However, we can assume that psychrophilic bacteria, which might grow well in air conditioning systems, are more than mesophilic one during cooling season, as shown in the results for gyms and libraries buildings.
Done, we agree with reviewer’s observations; in the article we have underlined the usefulness and importance of the indices to detect changes in the environmental microbial biocenosis. The GIMC, IMC and AI indices are intended to provide a quantitative and qualitative measure of aerial microbiological contamination. The increase of these indexes can suggest the presence of potentially pathogenic, allergenic and/or immunotoxic microbial species, which may not be of easy identification, representing a potential risk of respiratory diseases for sensitive people.
-I am not sure if the duplicated species growing on the same agar with different incubating temperatures do not reduce reliability of GIMC.
GIMC, IMC and IA calculations were determined from counts on TSA culture medium since it is frequently used in the determination of airborne concentrations of viable microorganisms. Presently, no uniformity exists in the methods used for the measurement and evaluation of indoor air quality. Exposure to airborne microorganisms in indoor and outdoor environments may be due to relatively low or large concentrations of bacteria and fungi, some of which may be pathogenic. The determination of total bacteria and fungal counts seems a more practical approach in environmental monitoring, We want to reiterate that the aim of our research is to apply this method to various occupational environments to verify its efficacy and to provide data useful for constructing reference values to evaluate IAQ; infact, these indices could represent a method for making criteria used to evaluate indoor environmental quality more uniform (Sofuoglu, S.C. and Moschandreas, D.J., 2003, Indoor Air 13, 332-343).
Yours sincerely,
Pietro Grisoli
Reviewer 2 Report
This is an interesting study where the authors evaluated airborne mesophilic and psychrophilic bacteria (by adjusting lab incubation temperatures) in Italian offices, gyms, and libraries (n=10 for each) and used three indices of microbial contamination for reporting the data instead of reporting typical culturable microbial concentration values. The study is important because the findings are relevant for microbial exposures to large number of people in the test locations and might be useful for better understanding the causes of sick building syndrome in the future. One major problem in the study is, however, limited amount of data. The authors collected only one sample per month in triplicate (I think colony counts in three exposed agar plates were averaged and presented in the table) and just 10 data points are reported in the Tables. Because culturable microbial concentrations have significant diurnal and seasonal variations, this limited amount of data cannot represent actual airborne microbial load to which people in these offices, gyms, and libraries are exposed. Another problem is, only culturable microroganisms are considered in this study and this is now well-known that both culturable and non-culturable microorganisms are responsible for various respiratory health hazards and both can function as immunomodulators.
Specific comments:
1. Title: The title is slightly misleading. I am not sure if gyms can be considered as recreational environment or not. I think it would be better to specify that the study was conducted in offices, gyms, and libraries.
2. Tables: Please add another column on standard deviations of data in all tables.
3. Line 188-189: Please present the data collected from outdoors and calculate Indoor:Outdoor ratios. Present this data in a separate table for all locations. This information is vital in determining the sources of indoor biocontamiantion.
4. Line 242-243: How the AI index is important for detecting the accumulation and proliferation of microorganisms in ventilation systems or in buildings is still unclear. The authors may provide some more information with references.
Author Response
Reviewer 2 comments
This is an interesting study where the authors evaluated airborne mesophilic and psychrophilic bacteria (by adjusting lab incubation temperatures) in Italian offices, gyms, and libraries (n=10 for each) and used three indices of microbial contamination for reporting the data instead of reporting typical culturable microbial concentration values. The study is important because the findings are relevant for microbial exposures to large number of people in the test locations and might be useful for better understanding the causes of sick building syndrome in the future. One major problem in the study is, however, limited amount of data. The authors collected only one sample per month in triplicate (I think colony counts in three exposed agar plates were averaged and presented in the table) and just 10 data points are reported in the Tables. Because culturable microbial concentrations have significant diurnal and seasonal variations, this limited amount of data cannot represent actual airborne microbial load to which people in these offices, gyms, and libraries are exposed. Another problem is, only culturable microroganisms are considered in this study and this is now well-known that both culturable and non-culturable microorganisms are responsible for various respiratory health hazards and both can function as immunomodulators.
We agree with the Reviewer that the number of samples is not high but this was mainly due to the objective difficulty of sampling at different times of the day in some of the analyzed sites. However it should be remembered that sometimes the biological research works with small samples and stastistical tests may be applied. It is true the fact that the largest the sample size is, the smallest the variability becomes. But we can say that ‘large sample are more useful than small samples if the large is unbiased’ as suggested also by W.C. Schefler in ‘Statistics for the Biological Sciences’ (1979). We feel that the microbiological determinations performed, and the GIMC, IMC, IA indexes, may represent simple and useful tools for the measurement and evaluation of biological risks. However we agree with Referees that these kind of evaluations are still problematic and far from the possibility to certainly state the correlation of the IAQ with the building-related symptoms (BRS). To corroborate our thesis, let us refer to a commentary just published in Indoor Air 2003:13: “Indices for IEQ and building-related symptoms” by Mark J. Mendell (Indoor Environment Departement, Lawrence Berkeley National Laboratory) the author states:“Currently, we do not have sufficient knowledge about risks to specific indoor exposure to define acceptable indoor concentrations that prevent BRS”… “Action to protect the public health need not and should not wait for complete understanding of causal exposures and biological mechanisms, if effective health-protective actions can be identified” … “We need to define acceptable performance for IEQ indices and then conduct strong research studies to create and validate such indices. The process would begin with the construction of a set of biologically plausible metrics for an IEQ index, followed by iterative evaluations of metrics, both as multiple single indices and combined in various composite indices, in order to test their correlation with BRS.” Regarding the determination of culturable bacteria, “viable but non-culturable” bacteria can probably be found in the air. This phenomenon is currently being much studied in various natural environments (see, for example, H.C. Wong and P. Wang, JAM, 2004, 96, 359-366) and, as indicated in the discussion of the paper, it can be hypothesised that only a small fraction of airborne bacteria can be recovered from culture media. It is currently not possible to use methods to determine the “true total” bacterial counts.
1. Title: The title is slightly misleading. I am not sure if gyms can be considered as recreational environment or not. I think it would be better to specify that the study was conducted in offices, gyms, and libraries.
Done, we have changed the title to avoid misunderstandings.
2. Tables: Please add another column on standard deviations of data in all tables.
Done.
3. Line 188-189: Please present the data collected from outdoors and calculate Indoor:Outdoor ratios. Present this data in a separate table for all locations. This information is vital in determining the sources of indoor biocontamiantion.
Done, we have added a table with the contamination values detected outside the buildings in the three sampling periods. We have clarified the role of the IA index, which represents the ratio between the contamination values determined within buildings and those outside, as, already, specified in materials and methods of the paper.
4. Line 242-243: How the AI index is important for detecting the accumulation and proliferation of microorganisms in ventilation systems or in buildings is still unclear. The authors may provide some more information with references.
Done.
The Amplification Index (AI) describes global indoor aerial modification. Generally there are no relations between indoor and outdoor fungal and the bacterial counts. However, we do not want to limit our judgement of Indoor Air Quality (IAQ) only to the measurement of single parameters, but also to the global value of microbial contamination. High AI values may only be indicative of IAQ deterioration when caused by high total fungal and bacterial counts.
Yours sincerely,
Pietro Grisoli
Round 2
Reviewer 1 Report
I can’t find the corresponding information to the second question in the manuscript, though it was written “Done”.
The authors insist on the usefulness of the overall index of airborne microbial contamination throughout the manuscript and the answers to reviewer’s comments. However, we should understand what the resulting values indicate and then to take an action. In many countries, total airborne bacteria and fungi are sampled, cultivated and evaluated separately to evaluate indoor microbial contamination and to take an appropriate actions to each bacteria and fungi, because they are from different sources. The authors need to explain why we should use the GIMC instead of conventional separate colony counts as in ISO 16000 series and what we know more by using them. And as for the third question on IMC, the authors just explain the reason as an exception. It might be impossible to use the index without feasible reason of such exception case.
The title “Airborne microorganisms in offices, gyms, and libraries” is not specific and vague. I suggest more specific title, for example, “Application of airborne microorganism index to offices, gyms and libraries.”
Author Response
Reviewer 1
I can’t find the corresponding information to the second question in the manuscript, though it was written “Done”.
In the materials and methods we added the number of persons and the air speed, for the different types of environments. Other data are not available.
The authors insist on the usefulness of the overall index of airborne microbial contamination throughout the manuscript and the answers to reviewer’s comments. However, we should understand what the resulting values indicate and then to take an action. In many countries, total airborne bacteria and fungi are sampled, cultivated and evaluated separately to evaluate indoor microbial contamination and to take an appropriate actions to each bacteria and fungi, because they are from different sources. The authors need to explain why we should use the GIMC instead of conventional separate colony counts as in ISO 16000 series and what we know more by using them. And as for the third question on IMC, the authors just explain the reason as an exception. It might be impossible to use the index without feasible reason of such exception case.
The proposed indexes complete the information provided by the single classes of airborne microorganisms and together they allow the analysis of the indoor air quality (line 349-350).These indexes represent an integration regards the determination of the single microbial parameters, as already explained in the article (line 267-282). The GIMC describes global indoor aerial modification. This index helps to understand quickly if there has been a change in microbial biocenosis within the building compared, for example, to a previous sampling campaign. In fact, GIMC values may only be indicative of IAQ deterioration when caused by high total fungal and bacterial counts. The IMC indicates the presence of obligated mesophilic bacteria, organisms of probable anthropic origin. The index represents a useful tool in the determination of bacteria growth caused by hypoventilation and overcrowding. (IMC>1). The use of these indexes might therefore allow an analysis of the air-diffused microbial situation present in a particular structure or in different structures with the possibility of finding threshold values, to be used as a reference for the periodic control of the microbiological air quality of that environment and of other buildings with similar characteristics.
The title “Airborne microorganisms in offices, gyms, and libraries” is not specific and vague. I suggest more specific title, for example, “Application of airborne microorganism index to offices, gyms and libraries.”
Done. As required, we change the title even if the last version was suggested by the other Reviewer.
Reviewer 2 Report
The authors adequately responded previous critiques. The revised manuscript is significantly improved. The authors may correct following minor errors:
Table 1 and 3 titles: Please replace ‘counts’ with ‘concentrations’.
Line 317: Add space between In and fact.
Line 329: Delete ‘it’.
Author Response
The authors adequately responded previous critiques. The revised manuscript is significantly improved. The authors may correct following minor errors:
Table 1 and 3 titles: Please replace ‘counts’ with ‘concentrations’.
Done.
Line 317: Add space between In and fact.
Done.
Line 329: Delete ‘it’.
Done.